# Issues and Opportunities Associated with Trophy Hunting and Tourism in Khunjerab National Park, Northern Pakistan

**DOI:** 10.3390/ani10040597

**Published:** 2020-04-01

**Authors:** Wajid Rashid, Jianbin Shi, Inam ur Rahim, Shikui Dong, Hameeda Sultan

**Affiliations:** 1School of Environment, Beijing Normal University, Beijing 100875, China; wajid@uswat.edu.pk (W.R.); dongshikui@sina.com (S.D.); 201839180002@mail.bnu.edu.cn (H.S.); 2Department of Environmental and Conservation Sciences, University of Swat, Mingora 19130, Pakistan; 3Centre for Applied Policy Research in Livestock (CAPRIL), Department of Climate Change and Livestock, University of Veterinary and Animal Sciences, Lahore 54000, Pakistan; inam.rahim@uvas.edu.pk

**Keywords:** trophy hunting, mass tourism, Pamir, eco-tourism, human–Snow leopard conflict

## Abstract

**Simple Summary:**

Trophy hunting and mass tourism were introduced to Khunjerab National Park, northern Pakistan to generate income for the community and help conserve and sustain the ecosystem in the region. These initiatives have provided economic benefits, but only at the cost of other environmental problems, as both trophy hunting and mass tourism have resulted in various ecological issues. Trophy hunting has not been based on scientific population data and has thus not helped increase numbers of wild ungulates or wild carnivores. Although mass tourism has increased enormously in this region, it has damaged the ecosystem through pollution generation and negatively impacted wildlife. We suggest that trophy hunting should be stopped, and mass tourism should be shifted to ecotourism as a sustainable solution to help improve the ecosystem, while generating income for the local community. Further studies are required to investigate ecotourism as a potential mitigation measure for the conservation issues in this region.

**Abstract:**

Trophy hunting and mass tourism are the two major interventions designed to provide various socioeconomic and ecological benefits at the local and regional levels. However, these interventions have raised some serious concerns that need to be addressed. This study was conducted in Khunjerab National Park (KNP) with an aim to analyze comparatively the socioeconomic and ecological impacts of trophy hunting and mass tourism over the last three decades within the context of sustainability. Focus Group Discussions (FGDs) with key stakeholders and household interviews were conducted to collect data on trophy hunting and mass tourism, and on local attitudes towards these two interventions in and around KNP. The results revealed that 170 Ibex (*Capra sibirica*) and 12 Blue sheep (*Pseudois nayaur*) were hunted in the study area over the past three decades, and trophy hunting was not based on a sustainable harvest level. Trophy hunting on average generated USD 16,272 annual revenue, which was invested in community development. However, trophy hunting has greatly changed the attitudes of local residents towards wildlife: a positive attitude towards the wild ungulates and strongly negative attitude towards wild carnivores. In addition, trophy hunting has reduced the availability of ungulate prey species for Snow leopards (*Panthera uncia*), and consequently, Snow leopards have increased their predation on domestic livestock. This has, in turn, increased human–snow leopard conflict, as negative attitudes towards carnivores result in retaliatory killing of Snow leopards. Furthermore, according to official record data, the number of tourists to KNP has increased tremendously by 10,437.8%, from 1382 in 1999 to 145,633 in 2018. Mass tourism on average generated USD 33,904 annually and provided opportunities for locals to earn high incomes, but it caused damages to the environment and ecosystem in KNP through pollution generation and negative impacts on wildlife. Considering the limited benefits and significant problems created by trophy hunting and mass tourism, we suggest trophy hunting should be stopped and mass tourism should be shifted to ecotourism in and around KNP. Ecotourism could mitigate human–Snow leopard conflicts and help conserve the fragile ecosystem, while generating enough revenue incentives for the community to protect biodiversity and compensate for livestock depredation losses to Snow leopards. Our results may have implications for management of trophy hunting and mass tourism in other similar regions that deserve further investigation.

## 1. Introduction

Trophy hunting programs, if designed and implemented scientifically and appropriately, can provide various socioeconomic and ecological benefits at the regional and global levels. However, trophy hunting has emerged as a debatable and hot issue in the world due to its profound socioeconomic and ecological consequences [1,2,3]. The issues and opportunities associated with trophy hunting are varying in different geographical settings [4]. There have been many cases in which trophy hunting can generate income for the local community [5,6], conservation benefits in some countries [1,6], and financial support for re-introduction of wildlife in some regions [7].

In some cases, trophy hunting generates few benefits for the local community [8,9], and income generated by trophy hunting is not distributed appropriately to the community helping in conservation of trophy animals [10,11]. A common problem with trophy hunting is lack of monitoring of trophy wildlife populations [12], resulting in improper population estimates [13] and improper quota for trophy hunting [14,15]. There are other problems related to trophy hunting, including no appropriate permits [16], trophy size being not related to its price [17], and lack of restrictions on the age of trophy animals [12]. The consequences of trophy hunting include tilted population sex ratio due to selective harvesting of large males [18,19], changing population dynamics [20], decline of target trophy animal populations [21,22,23], loss of genetic diversity and genetic changes [24,25,26], and increasing human-wildlife conflicts [27,28]. All of these can ultimately result in the loss of wildlife species [29,30].

Like trophy hunting programs, mass tourism, if not managed and conducted appropriately, has profound negative ecological impacts, as it may destabilize the ecosystems and increase pollution [31,32]. Due to the associated negative environmental impacts including degradation of natural habitats and pollution, mass tourism is considered as an unsustainable activity [33,34,35].

In contrast to mass tourism, ecotourism programs could provide dual benefits, including biodiversity conservation and economic benefits for the community [36,37]. If designed and implemented wisely, ecotourism can serve as a source of funding for biodiversity conservation [38]. Strikingly, ecotourism is a non-consumptive activity and far more acceptable compared to trophy hunting [39]. Similarly, in areas that have tourism potential, e.g., scenic attractions with easy accessibility [40], trophy hunting may cause wildlife to flee or avoid human presence, thereby negatively affecting wildlife watching and ecotourism [41].

Khunjerab National Park (KNP) was notified in 1975 by the government of Pakistan with an area of 4455 km^2^ to protect Marco polo sheep (*Ovis ammon polii*) and Snow leopard (*Panthera uncia*) in northern Pakistan. KNP is a combination of two different regions: one region is owned by communities represented by Khunjerab Village Organization (KVO) and named as KVO’s KNP; the other one is owned by Shimshal Community and named as Shimshal KNP [42,43]. The Shimshal’s KNP covers 3/4 of the national park area, but is not managed properly with almost no presence of the national park staff. There are no road accesses to Shimshal’s KNP and negligible tourism, and only limited adventurous trekkers visit there. On the other hand, KVO’s KNP is the managed part of the national park with officials being permanently employed for guarding and management inside this part. KVO’s KNP is only one fourth (1/4) of the total area of KNP, but almost all of the tourism in KNP is concentrated in this region [44].

Trophy hunting was introduced in 1993 into the Community Controlled Hunting Area (CCHA) that lies in the buffer zone of KNP [45] and is managed by KVO (so it is referred to as KVO’s CCHA). The aim was to increase the population sizes of prey ungulates including Ibex (*Capra sibirica*) and Blue sheep (*Pseudois nayaur*) for Snow leopards, and to generate sustainable income for the community at the same time [46]. An entrance fee was imposed for tourists to KNP since 1999. There have been few studies on the issues and impacts of trophy hunting and mass tourism in KNP, though trophy hunting and mass tourism have been operated there for several decades. Therefore, the current study was conducted to: (1) analyze the trend of trophy hunting and mass tourism in KNP; (2) identify and explore impacts of trophy hunting and mass tourism on conservation and local communities; and (3) recommend potential solutions to address the issues concerned with trophy hunting and tourism in KNP. Our results may have implications for better management of trophy hunting and mass tourism in other similar regions.

## 2. Materials and Methods

### 2.1. Study Area

The study area with a total area of 2407 km^2^, consisting of two portions—KVO’s KNP inside the national park and KVO’s CCHA in the buffer zone of the national park, lies between 36°33′ N to 37°01′ N and 74°47′ E to 75°26′ E in Hunza District of northern Pakistan (Figure 1). The study area has an altitudinal range from 2439 to over 4880 m above sea level [43,47], and is bounded in the North and North-east by Xinjiang Province of China, in the east and south by Shimshal Valley and Shimshal CCHA. Misgar CCHA and Chipursan CCHA lie in the west [42].

The KVO’s KNP has traditionally been used by the community for grazing livestock with an area of 1168 km^2^, but it is now managed by the Parks and Wildlife Department (PWD) for its resources and mainly used for tourism to generate revenues. On the other hand, KVO’s CCHA with an area of 1239 km^2^ is managed by KVO for trophy hunting of Ibex and blue sheep. There are no natural or man-made barriers between the two portions of the study area to prevent wild animals from moving into either portion [43,48].

The annual precipitation in the study area ranges from 200 mm to 900 mm and is mostly received in winter in the form of snow. The average temperature is below 0 °C from October onward and rises to about 27 °C in May [49]. The key mammal species in KNP are Siberian ibex (*Capra sibirica*), Blue sheep, Marco polo sheep, Snow leopard, Wolf (*Canis lupus*), Brown bear (*Ursus arctos isabellinus*), Lynx (*Felis lynx*), Tibetan red fox (*Vulpes vulpes montana*), Golden marmot (*Marmota caudata aurea*) and Cape hare (*Lepus Capensis*) [42].

KVO is a community-based organization of the seven villages in the buffer zone of KNP, and is responsible for conservation programs in the region, including trophy hunting. KVO receives 80% of the trophy hunting revenue and also 80% of the revenue from KNP’s entry fees [50].

### 2.2. Data Collection

Qualitative and quantitative approaches were used to collect data on the aforementioned parameters. Particularly, we used Focus Group Discussions (FGDs) and household interviews for data collection during 2018 with well-developed questionnaires. FGD is commonly used as a tool to collect scientific data from individuals or communities about a specific topic [51,52] and is also employed in conservation sciences. Furthermore, FGD is mostly used for evaluation of management interventions or policies which are already in place [53,54,55]. FGD tool was selected for this study for the following reasons. Firstly, it provides the different stakeholders (local community, National Park staff and KVO’s staff) an opportunity to discuss the conservation issues in a non-instructive manner. Secondly, FGD is flexible in allowing the discussion to progress gradually, thus prompting sharing of information and perspectives. Thirdly, FGD can dig out in-depth data about conservation issues. Finally, FGD can enrich the data obtained from household interviews and different records (including mass tourism and trophy hunting data) with additional information which is more representative of the ground situation.

Thus, a total of four FGDs were conducted: one with KVO’s officials (including KVO’s President and all related staff); one with the national park officials (Park Ranger, watchers); and the remaining two with the community members that were conducted in the main villages of Morkhun and Sost centre (Appendix A). For the community FGDs, a total of 13 community members were selected from the two main villages (particularly the elders and learned).

The FGD with the KVO’s officials was conducted to assess their perception about trophy hunting and its emerging problems in KVO’s CCHA (Appendix A). Besides, records of trophy hunting in the past three decades including number of wild ungulates hunted (Appendix A) and revenue generated were obtained from KVO’s officials. FGD with the national park staff was conducted to assess their perceptions of tourism and its issues inside KVO’s KNP. Data on the status of tourism (e.g., number of visitors, tourism income, etc.) in the past 20 years in KVO’s KNP were provided by the national park staff (Appendix A). While FGDs were conducted with community members to assess the community perceptions towards trophy hunting, wild ungulates and wild carnivores, the stories of retaliatory killing events of Snow leopards were also recorded. Data on changes in accessibility and tourism to KVO’s KNP and depredation of livestock were obtained by questionnaire surveys of 106 households. These households were randomly selected from all the 317 households in the seven villages in the buffer zone of KNP.

## 3. Results

### 3.1. Conservation Impact of Trophy Hunting

The FGDs revealed the perceptions of local community members regarding trophy hunting. Trophy hunting has changed the perspective of the local community towards wildlife in an antagonistic way.

#### 3.1.1. Perception about the Wild Trophy Ungulates

All of the participants (100%) in the community FGDs agreed that the number of the trophy ungulates (both Ibex and Blue sheep) should increase in the area (Table 1). They viewed the economic benefits from trophy hunting as the main reason for the positive attitude in the community.

#### 3.1.2. Perception about the Wild Carnivores in the Community

A majority (69%) of the community participants had negative attitude towards Snow leopards. They wanted the number of Snow leopards to decrease. About 77% of the FGD members perceived Snow leopard as a pest in the trophy hunting area (Table 1). Similarly, the community showed a strongly negative attitude towards wolves, as shown by the FGD result that all of the community FGD participants (100%) would like the number of wolves to decrease.

#### 3.1.3. Community Perception about the Impact of Trophy Hunting on Retaliatory Killing of Snow Leopards

Human–Snow leopard conflict was prevailing in the study area. Snow leopards were killed due to their depredation on livestock, and more importantly due to their depredation on the wild trophy ungulates. The FGDs with the community revealed that 13 Snow leopards were reportedly killed in the study area from 2011 to 2018 (Table 2). Snow leopard killings were not reported outside this area, because local community members feared being punished by the government. Furthermore, all of the interviewed community members in the FGDs told us the retaliatory killings of Snow leopards would continue unless there was a comprehensive mechanism to compensate for the losses of livestock due to depredation by Snow leopards.

#### 3.1.4. Opinions about Continuity of Trophy Hunting in the Future

The FGDs with the community and the KVO officials revealed that all of the participants agreed that trophy hunting had generated economic benefits for the community as a whole. Interestingly, only the community participants elaborated that trophy hunting should be continued for the economic benefits to community in KVO’s CCHA, but all the KVO’s officials openly admitted that trophy hunting was not a conservation tool in KVO’s CCHA.

*One of the problems with trophy hunting in KVO’s CCHA was the lack of population census or survey of the trophy ungulates in recent years and the annual quotas for trophy hunting were estimated subjectively. Additionally, the required age of a single trophy ungulate should be about 12 years old, but most trophy animals were much younger than this. The population sizes of trophy ungulates were decreasing at a faster rate in recent years, so the trophy hunting program in KVO’s CCHA was unsustainable. Proper scientific surveys were required for determining the number of trophy animals to be harvested in a single year. Moreover, trophy hunting in KVO’s CCHA should be stopped at the earliest for the benefits of wild ungulates. Instead, some conservation-friendly alternative to trophy hunting was desired*.(FGD with KVO’s Officials)

#### 3.1.5. Livestock Holdings and Livestock Depredation 

The questionnaire survey of the 106 sampled households revealed that the community in the study area was dependent on livestock for their livelihood. Livestock population data showed that sheep (808) were the dominant type of livestock in the study area, followed by goats (677) and cows (305). Yak (154) was kept in small number in the study area. In the last year, 52 sheep, 47 goats, and 18 yaks were depredated. Cows were depredated less by comparison, with the least number (8) last year (Table 3).

### 3.2. Economic Returns of Trophy Hunting Program in KVO’s CCHA

Trophy hunting of Ibex and Blue sheep were introduced into KVO’s CCHA in 1993 and 2004, respectively. A trophy hunting season in KVO’s CCHA starts in early October and ends in late March.

A total of 170 Iibex were legally hunted as trophies in the past 26 years. The number of Ibex hunted per year remained only one in the first several years from 1993, but it generally increased since 2000 (Figure 2A). Twenty-nine Ibex were hunted in the hunting season of 2016–2017, while the number of Ibex hunted almost halved (14) in 2017–2018 (Figure 2A).

Trophy hunting of Ibex in CCHA had generated a total of US$326,963 since its inception in 1993 (Figure 2B). The proceeds were shared between KVO and the government with 75% of the proceeds going to KVO before 2000–2001 and 80% after 2000–2001. The total share from trophy hunting of Ibex for KVO was US$261,321, accounting for 79.92% of the total earnings.

The charge for hunting a single Ibex was different for Pakistani and foreign hunters. Initially, the charge was much low for Pakistani hunters, but it generally increased since the 1990s (Figure 2C). The charge for hunting one Ibex by foreign hunters was higher than by Pakistani hunters in most years, except in 2007–2008, when the charges were the same. There was more variation in the charge for foreign hunters than for the Pakistani hunters across the entire study period.

A total of 12 adult male Blue sheep were hunted since 2004, among which 10 males were hunted by foreign hunters. No single Blue sheep was hunted in eight years since 2004 (Figure 3A).

Like the charge for trophy hunting of Ibex, the charges for trophy hunting Blue sheep were different for Pakistani hunters and foreign hunters in most of the years (Figure 3B). Pakistani hunters paid less (US$5000) than foreign hunters (US$8600) in 2014. The charge reached the highest (US$15,000) in 2007, but reduced to almost half in the following years. The decreased charge for trophy hunting of the endangered Blue sheep could be attributed to its lower charge in the nearby valley of Shimshal, where there were more Blue sheep than in KVO’s CCHA. The reduced charge for Blue sheep in Shimshal drove down the charge in KVO’s CCHA.

Trophy hunting of Blue sheep had generated a gross revenue of US$96,100 in the past 14 years. KVO’s share was US$76,880 (80%), and the government share was US$19,220 (20%).

### 3.3. Mass Tourism and Its Economic Benefits

KVO’s KNP has become a major tourist attraction in Pakistan in recent years. The number of tourists to KVO’s KNP has fluctuated widely, but has generally increased in recent years, generating economic benefits correspondingly (Figure 4a). Only 1382 people visited KVO’s KNP in 1999, while the number of tourists surged by 10,437.8% to 145,633 in 2018. There was a sharp decrease in the number of tourists (13,331) in 2002 due to the occupation of neighboring Afghanistan by US troops after the September 11 attacks in the United States of America in 2001. Similarly, in 2005, the Kashmir earthquake resulted in another decreased flow of tourists (14,102) to KVO’s KNP. The most significant decline in the number of tourists (4,644) took place in 2010 due to the Attaabad landslide (damming Hunza River) which resulted in a portion of the road being submerged by the Attaabad Lake. In recent years, the number of tourists to KVO’s KNP has increased by many folds, partially due to construction of the China-Pakistan Economic Corridor, which has highlighted this important region (Figure 4b).

However, the many-fold increased number of tourists did not correspondingly increase revenue. More than half a million (678,075) tourists entered into KVO’s KNP in the past 19 years, but a revenue of only US$457,137 (US$365,710 for community share and US$91,427 for government share) was generated. Even in 2018 when the number of tourists to KVO’s KNP surged to a record-high (145,633), tourism revenue was only US$101,895 (Figure 4b). This was mainly because the admission charge to KVO’s KNP was low: less than US$1 for Pakistani tourists since the beginning (1999) and US$4 for foreign tourists between 1999 and 2011. The entry fee was PKR 20 (Pakistani Rupees) (on average equal to US$0.3) for domestic tourists, and US$4 per person for foreign tourists between 1999 and 2010. The entry fee was raised to PKR 40 (on average equal to US$0.4) between 2011 and 2016 and then to PKR 100 (on average equal to US$0.8) since 2017 for domestic tourists (Figure 4c). The entry fee for foreign tourists was raised to US$8 per person from 2011 onward. Despite the increased entry fees per foreign tourists, tourism revenue did not increase proportionally because most tourists were from Pakistan.

### 3.4. Comparison between Trophy Hunting and Mass Tourism

The revenue from trophy hunting of Ibex and Blue sheep during the period (1993–2018) was calculated to be US$423,063, while that from mass tourism during the period (1999–2018) was US$457,137 in the study area. Although the entry fee to KVO’s KNP was imposed from 1999, tourism to KVO’s KNP generated a higher and more reliable revenue stream than trophy hunting (Figure 5). Trophy hunting returns were even zero in some years due to unavailability of trophy hunters or the absence of trophy animals in KVO’s CCHA, whereas tourism grew steadily and dramatically after 2010 (Figure 5). The number of tourists to KNP increased from 2017 to 2018, but tourism revenue (in US$) decreased in 2018 due to devaluation in the national currency against US dollar (Figure 4c).

### 3.5. Perception of National Park Staff about Mass Tourism Inside KVO’s KNP

All of the national park staff participating in FGDs observed that mass tourism was increasing at a faster rate in recent years. As evidence, they pointed out tourists even visited the national park in the winter season in recent years. They revealed that a proposal was developed to open the Khunjerab Pass (connecting Pakistan and China) year-round for vehicles in the near future, whereas this high mountain pass was closed in winter due to heavy snowfall now. The opening of this trade corridor to winter traffic will be another boost to the tourism inside the national park.

Meanwhile, they expressed their concerns over mass tourism because mass tourism might have caused some eco-environmental problems within the national park.

*An estimated 4 Ibex and some other animals were killed per year in collisions with vehicles inside KVO’s KNP due to the lack of protection/fence on both sides of the road. Mass tourism has also resulted in pollution inside KVO’s KNP, which could be observed along the main road. The solid wastes are impacting the serene environment inside KVO’s KNP, and wild animals at times consume these wastes. In addition, noise from traffic disturbs wildlife inside the park*.(FGD with national park officials)

The KVO organized seasonal campaigns with volunteers to clean the environment inside the park (only in Khunjerab top), but these efforts were not enough to control the pollution problem as the area was very large and the mass tourism was increasing exponentially inside the park on a yearly basis.

### 3.6. Impact of Infrastructure on Tourism Inside the Park

The questionnaire survey (N = 106) showed that an overwhelming majority (97.2%) of the respondents believed accessibility to KVO’s KNP increased in the last 15 years, while only 2.8% of the respondents thought that accessibility did not increase. The increased accessibility to KVO’s KNP was mostly attributed to road construction and opening of the Attabad tunnels, while very few respondents believed that the improved security situation in the country contributed to the increased accessibility to KVO’s KNP (Figure 6).

## 4. Discussion

### 4.1. Trophy Hunting: Its Impacts on Trophy Ungulates

As currently practiced and as revealed by our study, trophy hunting does not seem to be a sustainable conservation tool for the ungulate species in our study area because it may cause a decrease in their population sizes. To sustainably hunt wildlife, a specific population census must be conducted with proper scientific methods. However, no such census has been carried out regularly in the study area for many years [46,56]. Appropriate monitoring of the trophy ungulate populations employing standard methodologies were previously proposed by studies in this area [50,57], but consecutive seasonal surveys are still lacking, which is a basic requirement of trophy hunting. Before 2006, proper population surveys involving the Wildlife department, local communities and NGOs were conducted (FGD with KVO’s Officials). The lack of regular population census of wild trophy ungulates has resulted in trophy hunting quotas that are based on guesstimates and often above the sustainable harvest level. Furthermore, most trophy hunters opt to hunt in KVO’s CCHA, possibly due to its easy access from the main Karakoram Highway and the perceived high conservation value of KNP [45]. Therefore, higher quotas (for the same year) have been set by the provincial wildlife department for Ibex trophy hunting in KVO’s CCHA, to meet the higher demand from the trophy hunters. This has caused trophy hunting of Ibex to exceed its annual quota in KVO’s CCHA.

A former study estimated the average density of Ibex to be 0.276 km^2^ in different valleys inside the national park in 2016 [42]. We used this estimate as a baseline and extrapolated it to KVO’s CCHA with an area of 1239 km^2^ and outside of the national park. The very optimistic estimated population size of Ibex was 342 in KVO’s CCHA in 2016. In Pakistan, previous studies [57,58,59] have used a trophy hunting quota allocation of 1–2% of a target population of similar wild ungulates for sustainable harvesting. With our optimistic estimate of 342 individual Ibex in KVO’s CCHA, about 3–6 Ibex trophies (1–2% of the population size) could be harvested sustainably in 2016. However, the current study revealed that 29 Ibex trophies were harvested in the single year of 2016, which was more than 6 times above the sustainable harvest quotas. A previous study [45] revealed that Ibex trophy hunting in 2006 was above 83.3% of the allocated quota for that year, i.e., about 11 Ibex were hunted against the allocated quota of 6 Ibex in KVO’s CCHA, which was set without any scientific basis. Trophy hunting without scientifically based harvest quotas may cause the Ibex population to crash in long term [45].

The study area is the westernmost part of the global Blue sheep distribution [46]. Previous studies in Pakistan have proposed that trophy hunting in similar wild ungulates may not be permitted if the total population is below 100 individuals [57,58]. There were about only 70 Bblue sheep in KVO’s CCHA in 2006 [45], suggesting the population was too low for sustainable trophy hunting of even a single Blue sheep. However, our study revealed that one Blue sheep trophy was harvested in 2006 and another two in 2007. A recent survey observed a total population of 104 Blue sheep in KVO’s CCHA in 2014 [46], but two trophy Blue sheep were harvested in that year. This shows that trophy hunting of Blue sheep is not sustainable in this area. Additionally, as the pre-requisites for Markhor (*Capra falconeri falconeri*) trophy hunting in this region, two trophy size males could be harvested if the total population is 150 individuals and 8 trophy size males were observed in consecutive winter seasons [50]. Unfortunately, consecutive winter seasons surveys are not in practice to find harvest quota in the study region employing this criteria.

The unsustainable trophy hunting of wild ungulates in an area may pose a threat to their survival in the long run [60]. This issue was also stressed by the KVO officials who considered that current trophy hunting in the KVO’s CCHA was not sustainable, and it was not helping in conservation of wild ungulates. Studies conducted previously have pointed out that proper scientific population censuses are rare [13,46,61]. Efforts must be undertaken to assess and monitor population trends at an appropriate time interval to determine if wildlife populations are declining on account of trophy hunting [46,62,63].

In contrast to our study, trophy hunting of Suleiman Markhor (*Capra falconeri jerdoni*) and Afghan Urial (*Ovis orientalis cycloceros*) in the Balochistan province of Pakistan has been carried out sustainably since 1986. Trophy hunting in that region has been conducted on the basis of regular and scientific surveys of wild ungulate populations and has resulted in considerable economic benefits to the community and helped in the restoration of these wild ungulates [6,64].

### 4.2. Trophy Hunting: Its Impacts on Carnivores

Trophy hunting in the study region is only centered around trophy ungulates because economic value is attached to trophy ungulates only. Previous study has shown that trophy hunting has not helped in changing negative perception against wildlife in a positive way [63]. In our study, the local community’s perceptions about wolves were strongly negative, and all of the community members wanted to decrease or eliminate the wolf population. However, community’s perceptions about Snow leopards were less negative compared to wolves, possibly due to conservation programs or awareness-raising in the community about Snow leopards. A study in Mongolia found a similar situation, where the community had stronger negative perceptions on wolves than on Snow leopards [65].

Snow leopards are the apex predator, largely depending upon wild ungulates as their prey species. The majority of local community residents perceive Snow leopards as their enemy. Human-Snow-leopard conflict has increased with the introduction of trophy hunting. Studies have shown that, after the introduction of trophy hunting, local people have two reasons to kill Snow leopards: predation on trophy ungulates and depredation on livestock [27,66]. Our study reveals that Snow leopards are often viewed as a threat to not only the livestock, but also trophy hunting because they are perceived to deplete wild ungulates, which local people believe would otherwise generate more trophy hunting revenues. The depredation of livestock by Snow leopard is high in our study area with about 12% of the yak, 7% of the goats and 6% of the sheep being depredated in last year. The trophy hunting program has decreased the number of wild ungulates, compelling the carnivores to depredate the livestock in this area. Due to unavailability of natural prey, Snow leopards are depredating more on livestock [67].

Our study shows that on average more than one Snow leopard was reportedly killed per year in the study area. However, a previous study has concluded that many local people have a negative perception on Snow leopards, and even kill them without reporting to the authority due to fear of penalty from the government [68]. Thus, it is more likely that the number of Snow leopard killings is higher than the number currently reported.

Trophy hunting may provide funds and incentives to protect wild ungulates in this region, but the benefits certainly do not extend to other wildlife species, even to rare carnivores like Snow leopards. Interestingly, trophy hunting has increased the perceived value of specific trophy animals [69], but the associated wild carnivores which prey on them are not valued [70]. Realistically, however, these carnivores also need to be conserved to ensure a continuing supply of ecosystem services and maintenance of ecosystem functions [71].

### 4.3. Trophy Hunting: Its Economic Benefits and Future

Trophy hunting in the KVO’s CCHA has generated a total revenue of US$423,063 from 1993 to 2018, which on average is equivalent to an annual revenue of US$16,271. The local community obtained on average 80% of the revenue, i.e., about US$13,000 per year. This seems to be a substantial income for the local community, given the exchange rate. However, the economic return from trophy hunting is expected to fluctuate and decrease due to the current unsustainable level of harvest of wild ungulates in KVO’s CCHA.

There is a big difference in the charges for Pakistani and foreign hunters to hunt either Ibex or Blue sheep in the same year. Pakistani hunters usually pay astonishingly lower prices to trophy hunt an endangered animal (e.g., Ibex). Interestingly, about 28.8% of all the Ibex trophies in the last four years were hunted by Pakistani hunters in KVO’s CCHA, and these hunters paid less than US$2000 per animal on average. As such, even though more Ibex are hunted than in previous years, there has not been a corresponding increase in trophy hunting revenues.

Trophy hunting has put a monetary value on ungulates species. The community has accepted trophy hunting due to the economic benefits it provides. Yet conflicted perceptions about the continuation of the trophy hunting were identified in the study area. Community members only wanted trophy hunting to be continued for its economic benefits. However, the KVO (although directly receiving the trophy hunting revenue) officials identified ecological problems caused by trophy hunting. They proposed that trophy hunting should be discontinued due to its negative ecological impacts. Trophy hunting resulted in monetizing wild trophy ungulates, but recent studies show less effect of population increase (especially the blue sheep), despite continuous trophy hunting for a few decades in these valleys. The total number of Blue sheep in KVO’s CCHA previously was 70 in 2006 [45] and 104 in 2014 [46] with no current estimates. Appropriate number of trophy size males in ungulates population is essential for successful trophy hunting programs and higher fecundity [46]. However, until now no study documented the exact number of trophy size males in the Blue sheep population in this area. The KVO officials revealed that the number of trophy size ungulates was decreasing at a faster rate in recent years. The ratio of female to young in the Blue sheep population was 1:0.5 in 2014 [46], representing a very low level of offspring and lower fecundity. Similarly, previous studies have shown that economic valuation and commodification of wildlife animals have created problems for conservation [72]. The issue with trophy hunting in many regions is that economic values in trophy hunting are asymmetric, i.e., wild trophy ungulates are economically valued, while wild carnivores are not valued, thereby resulting in conflict situations [73]. Instead, the Snow leopards are regarded as pests and killed in retaliation. Many studies have found out that giving monetary values to wildlife can result in conservation conflicts, whereas stressing the intrinsic values of wildlife helps in conservation [74,75]. Additionally, the monetization of biodiversity is not applied appropriately, thus not helping in conservation [76]. Some scholars have suggested it is our moral obligation to conserve wildlife, not just because of their monetary value [77,78]. Although trophy hunting could generate some income for the local community, the economic benefits cannot be the sole justification for trophy hunting to continue in the future [29,79]. 

### 4.4. Mass Tourism: Its Economic Benefits, Environmental Problems and Future

Mass tourism in KVO’s KNP has increased by 10,437.8% and generated an average revenue of US$33,904 per year. The local community obtained on average 80% of the total revenue, i.e., on average about US$27123 per year. For the local community having a total population of 317 households, this is a considerable income that can be used to improve their livelihood. Besides, as perceived by the national park staff in this study, the revenue from tourism will increase further in the coming years due to the increasing number of tourists into this national park. Such perceptions are similar to studies in other regions [80,81,82].

Compared to trophy hunting, tourism provided a more reliable and sustainable stream of revenue because the income from trophy hunting (US$16,271/year) was highly variable, partly due to non-availability of trophy hunters or non-availability of trophy size animals. As a result, no revenue was generated by trophy hunting in some years. On the other hand, the revenue from tourism (US$33,904/year) was sustainable and increasing year to year in KVO’s KNP.

The improved maintenance of highways, and construction of the Attabad Tunnels [83] and the China-Pakistan Economic Corridor (CPEC) passing through KVO’s KNP [84] have contributed to a tremendous increase in tourism, and this increasing trend is expected to continue in the future. Currently, the CPEC is helping to promote tourism in this region [85], and there is huge potential for mass tourism due to economic development, enhanced livelihood standards and improved infrastructure in the region. Previous studies have pointed out that tourism is generating the most sustainable flow of benefits to communities living with wildlife or near protected areas [86,87]. We thus believe that tourism will prevail in KVO’s KNP in the future and result in significantly increasing income for both KVO and the local communities.

Rapid increase in tourism could provide a good source of income [88,89], but uncontrolled tourism would also degrade the local environment [88,90]. Mass tourism has considerable negative environmental impacts and threatens local biodiversity, and thus ultimately can degrade the ecosystem [91,92]. The emissions from the vehicles transporting tourists are responsible for degrading the air quality [93] and the solid waste pollution generated by the mass tourism could cause degradation to environment and wildlife habitats [94]. Previous studies have revealed that stakeholders perceived environmental problems arose from tourism in an area [95,96,97]. In addition to the environmental problems, mass tourism also results in social problems over resource use [91,98]. As revealed by our study, the current tourism is adversely impacting the environment inside the KVO’s KNP due to the pollution and degradation of the habitat. These environmental challenges need a sustainable solution [91].

KVO receives 80% of the income from the entry fees to KVO’s KNP. The current income from the park entry fees is low for the local community because of the low entry fee charged for each tourist. There is huge potential and room to increase the income from tourism for the local community. The current entry fee (less than 1 US$ for local and 8 US$ for foreigner) to KVO’s KNP is significantly low in comparison to similar national parks in nearby countries. For example, the entrance fee to Kanha Tiger Reserve in India is US$14.80 per domestic tourist and US$47.47 per foreign tourist [99], and the entrance fee to Sagarmatha National Park in Nepal is US$30 per foreign tourist [100]. A very small increase in the entrance fee to KVO’s KNP could greatly increase income for the KVO community, with potential positive impacts on the national park and the entire community. In addition, studies in China, the United States and Malaysia have found that increasing the entry fee to a national park will result in a decrease in the total number of tourists [101,102,103]. This would potentially protect the KNP biodiversity from the adverse effects of mass tourism in the long run, while generating even more income than current tourism.

Ecotourism, in comparison to mass tourism, has more benefits [103] and could be a better option to balance conservation and local community development in the current context. Studies have shown that ecotourism is the best option for areas with wildlife availability and easy accessibility for tourists [40,80]. Previous studies have shown that ecotourism has dual benefits including environmental conservation and income generation for the community [104,105,106]. Eco-tourism could even replace trophy hunting as a sustainable revenue sources and a beneficial tool for conservation in this region [86] if we design and implement a better tourism plan.

## 5. Conclusions

Trophy hunting and mass tourism are providing economic benefits to the community in the buffer zone of KNP. Analyzing the success of these programs based only on income generation tells an incomplete story. Currently, both trophy hunting and mass tourism are failing to support conservation of wildlife in and around KNP. Trophy hunting is not beneficial to the trophy prey ungulates or to the endangered Snow leopards, which are killed in retaliation for their predation on wild trophy ungulates and livestock. Mass tourism, meanwhile, has damaging environmental consequences. KVO’s KNP has become a major tourist attraction, which could provide an alternative source of income to the community from entry fees. However, mass tourism has environmental consequences that need to be mitigated, possibly by shifting to ecotourism. Tourism revenue in KVO’s KNP could be increased many folds by raising the entrance fee slightly. Raising the entrance fee to a proper level could provide higher revenue for the community, and meanwhile potentially deter mass tourism, which could help restore degraded areas in the park.

Considering the limited benefits and significant problems created by trophy hunting and mass tourism, we suggest trophy hunting should be stopped in and around KNP. We conclude by recommending eco-tourism as a possible and sustainable alternative solution to address issues related to local community development and wildlife conservation, which deserves further investigation. Ecotourism could mitigate human–Snow leopard conflicts and help conserve the fragile ecosystem, while generating enough revenue incentives for the community to protect biodiversity and compensate for livestock depredation losses by Snow leopards. Ecotourism is considered to be economically viable, socially acceptable, ethically justifiable, and ecologically sustainable. Our results may have implications for management of trophy hunting programs and tourism in other similar regions. Future research can explore the possibility of ecotourism as an alternative to trophy hunting in easily accessible regions. 

## Figures and Tables

**Figure 1 animals-10-00597-f001:**
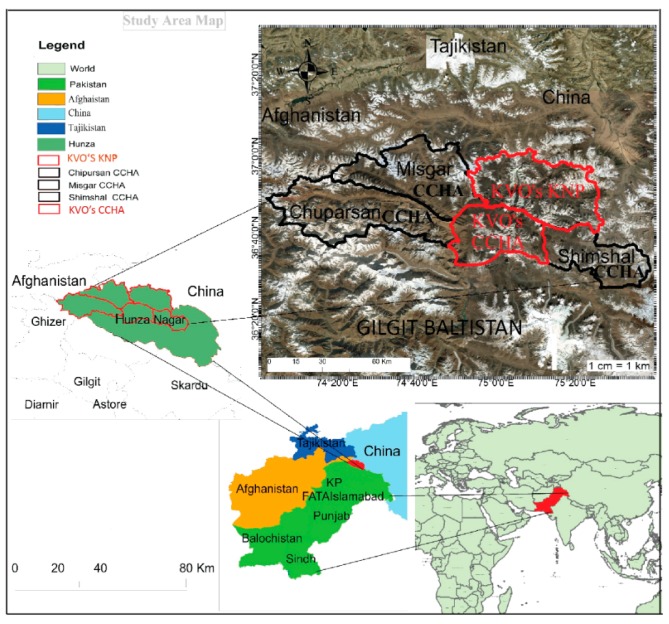
Map showing the study area consisting of Khunjerab Village Organization’s (KVO’s) Khunjerab National Park (KNP) and KVO’s Community Controlled Hunting Area (CCHA). The study area is located on Pakistan’s northern border with China.

**Figure 2 animals-10-00597-f002:**
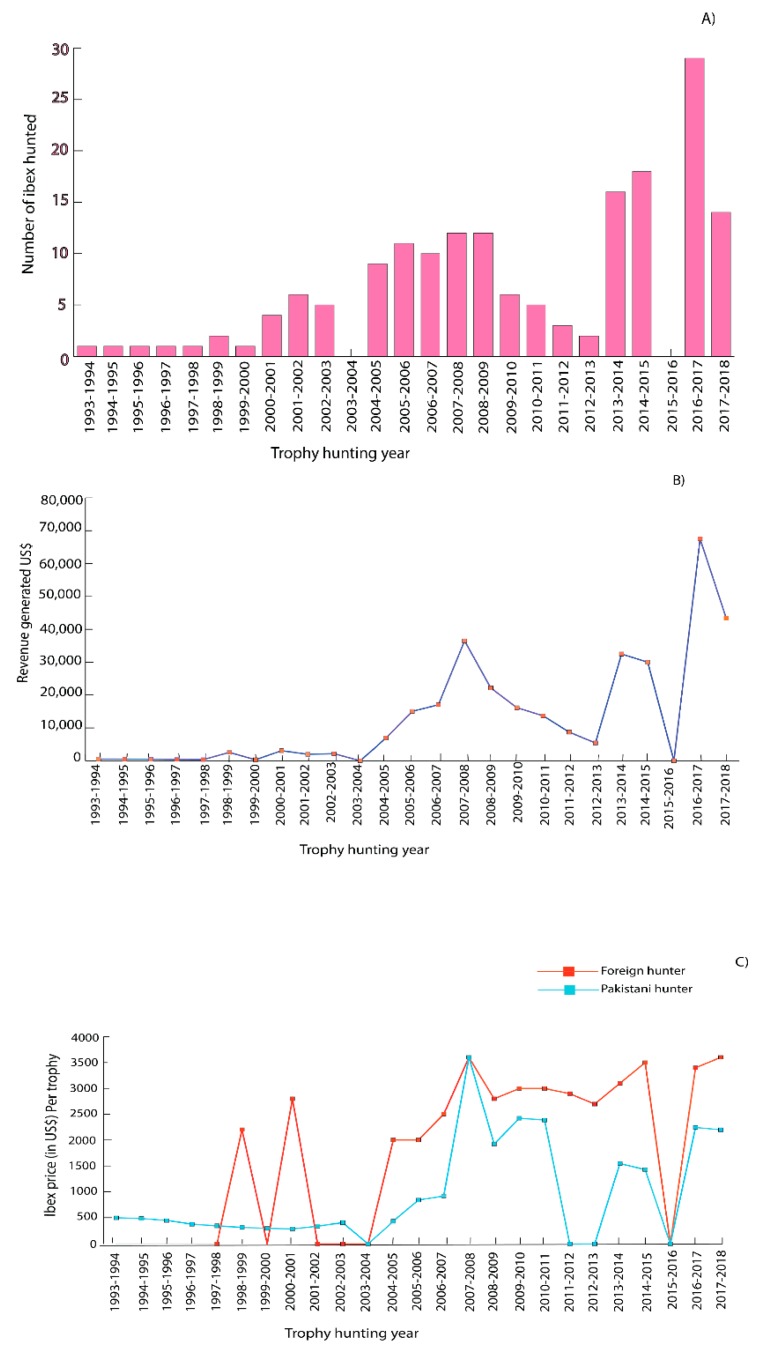
Trophy hunting of Ibex in KVO’s CCHA from 1993 to 2018: (**A**) The number of Ibex hunted in KVO’s CCHA; (**B**) Annual revenue generated from trophy hunting of Ibex; and (**C**) Charges for trophy hunting of an Ibex for a Pakistani hunter and foreign hunter from 1993 to 2018.

**Figure 3 animals-10-00597-f003:**
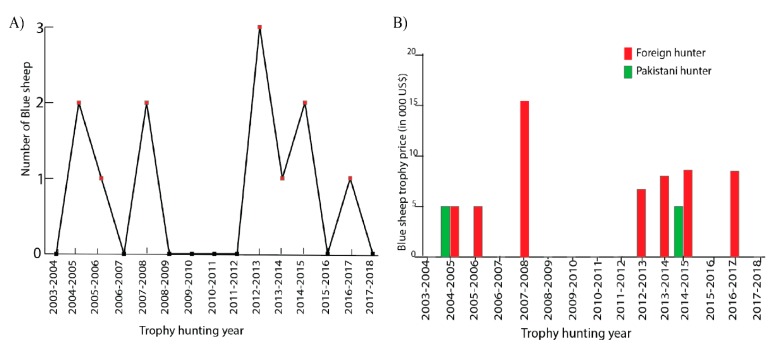
The trophy hunting of Blue sheep in KVO’s CCHA: (**A**) The number of Blue sheep hunted per year from 2004–2018; (**B**) Charges for trophy hunting a Blue sheep for a Pakistani and foreign hunter from 2004–2018.

**Figure 4 animals-10-00597-f004:**
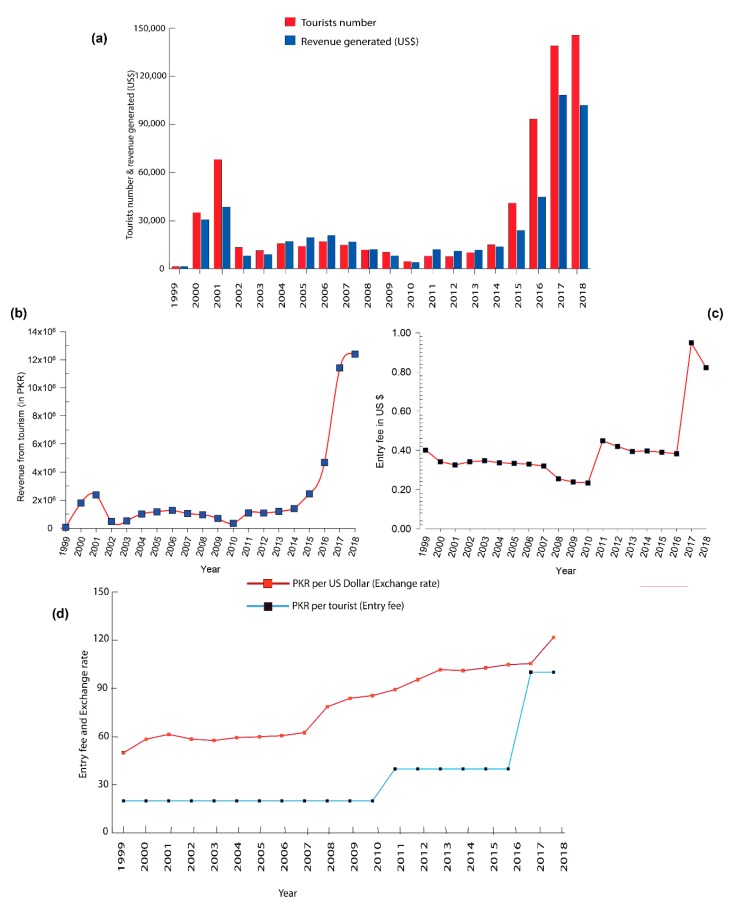
Tourism in KVO’s KNP: (**a**) Tourist number and revenue in US$ from 1999 to 2018; (**b**) Revenue in Pakistani Rupees (PKR) from 1999–2018; (**c**) Entry fee for a single tourist to KVO’s KNP in US$; and (**d**) Entry fee (PKR per tourist) and exchange rate (PKR per 1 US Dollar).

**Figure 5 animals-10-00597-f005:**
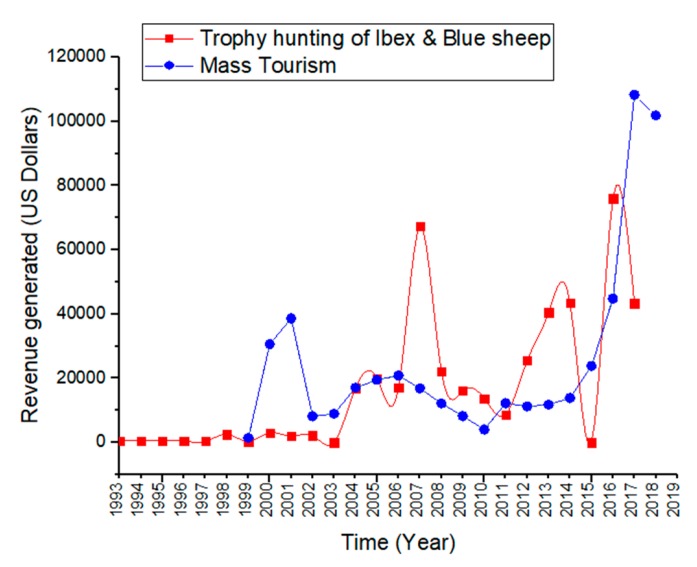
Comparison between trophy hunting (Ibex and Blue sheep) and tourism in the study area.

**Figure 6 animals-10-00597-f006:**
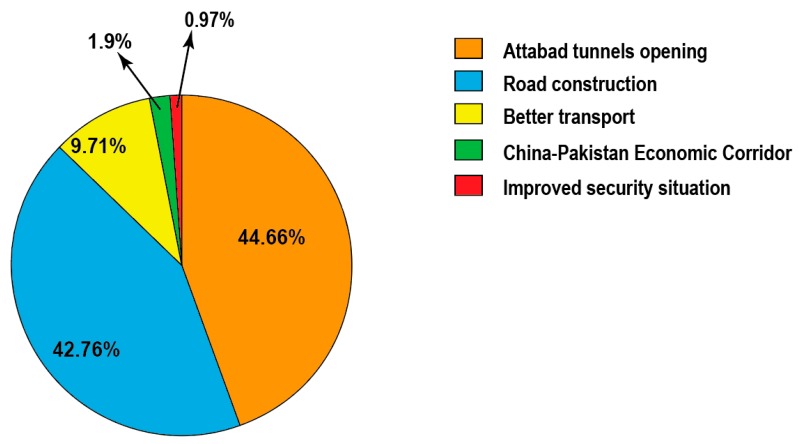
Respondents’ opinion on increasing accessibility to KVO’s KNP in the last 15 years. The perceived reasons were grouped together and shown in percentages (N = 106).

**Table 1 animals-10-00597-t001:** Demography and perceptions of the Focus Group Discussions participants

Category	KVO Officials FGD 1	National Park Staff FGD 2	Local Community			Total
			FGD 3	FGD 4	Sub-Total (FGD 3 & FGD 4)	
No. of participants	7	5	6	7	13	25
Age range (years)	37–55	25–43	28–48	30–67	–	–
Location of FGDs	Sost Bazar	Dhee post	Morkhun village	Sost village	–	–
Gender	All males	All males	All males	All males	–	–
Number of Ibex should increase	–	–	6/6	7/7	13/13	13/13
Number of Blue sheep should increase	–	–	6/6	7/7	13/13	13/13
Number of Snow leopards should decrease	–	–	4/6	5/7	9/13	9/13
Number of Wolves should decrease	–	–	6/6	7/7	13/13	13/13
Opinion about Snow leopards as pest animals in trophy hunting area	–	–	5/6	5/7	10/13	10/13
No compensation scheme for depredation loss	7/7	–	6/6	7/7	13/13	20/20
Benefits of trophy hunting	7/7	–	6/6	7/7	13/13	20/20
Disadvantages of trophy hunting	7/7		0/6	0/7	0/13	7/20
Problems due to mass tourism	7/7	5/5	–	–	–	12/12
Suggestion for Snow leopard conservation	Alternative conservation to replace trophy hunting	–	Compensation scheme	Insurance of livestock	–	–

**Table 2 animals-10-00597-t002:** Snow leopards reportedly killed in and around KNP from 2011 to 2018.

Month (Year)	Description	By Whom	Exact Location Where Snow Leopard(s) Was Killed
May or June 2011	One Snow leopard was killed in retaliation for mass killing of sheep and goats.	Livestock herders	Dhee Nala, Khunjerab National Park
2014	The hunters injured two Snow leopards. One died on spot, while the other died later.	Illegal hunters	Ghalapan, Khunjerab valley
2014	One Snow leopard was killed in retaliation	Farmers	Abgarch Khunjerab valley
February 2014	One Snow leopard was killed inside a livestock corral	Locals community	Yarzghirch, Chipursan, near Khunjerab National Park
March 2014	Two Snow leopards were killed inside livestock corral.	The locals	Kirman, Chipursan near Khunjerab National Park
23 June 2014	Four Snow leopards were killed by burning inside a cave. Petrol was used to make a flame and they were burned alive.	Locals	Boybur valley, Jamalabad, Khunjerab valley
December 2014	One Snow leopard was killed in Khuda abad near Sost area. The illegal hunter carried that Snow leopard and asked some locals to help in stuffing the Snow leopard hide.	Illegal hunter	Khuda abad, Sost, Khunjerab valley
2016	One Snow leopard was killed in retaliation for killing the livestock in the valley	Local farmers	Misgar Valley bordering Khunjerab National Park
**Total**	**13**		

**Table 3 animals-10-00597-t003:** Total number of livestock and number of livestock depredated in the study area last year.

Livestock Species	Number of Livestock (n)	Mean ± SD	No. of Livestock Depredated	Mean ± SD
Yak	154	1.5 ± 3.7	18	0.17 ± 0.5
Goats	677	6.4 ± 9.9	47	0.44 ± 1.3
Sheep	808	7.6 ± 8.8	52	0.49 ± 1.3
Cows	305	2.8 ± 1.6	8	0.08 ± 0.4
**Total**	**1944**	**–**	**125**	**–**

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
