# Peer review of "Issues and Opportunities Associated with Trophy Hunting and Tourism in Khunjerab National Park, Northern Pakistan"

_animals, 2020, doi:10.3390/ani10040597_

Round 1
Reviewer 1 Report
The revision has addressed my concerns from the previous review.
Author Response
Thank you very much.
Reviewer 2 Report
Paper is much improved and the authors have addressed my concerns. I think a more nuanced discussion of tourism forms is still possible but the paper is acceptable as it stands.
Author Response
Thank you very much.
This manuscript is a resubmission of an earlier submission. The following is a list of the peer review reports and author responses from that submission.
Round 1
Reviewer 1 Report
Review – Animals 240120
Trophy hunting in Pakistan
Overall this paper contains valuable information that is of very high local significance and global interest, but which does not offer entirely new globally innovative discussion on the topic. The data are valuable and come from a hard-to-reach context.
The paper requires extensive re-writing.
English is idiosyncratic and should be edited
the format for quotes used is incorrect – they should be in paragraphs indented if long, with form of source (eg community member) listed, or in quotes in text if short.
Introduction
Appropriate, but the detail of the National Park in footnote should be included in text in brief and offered in full in supplementary material
Methods
Reduce and make more concise the discussion of FGDs
Comment on the all male groups
Results
Could use some smaller quotes as well as longer paragraphs, and use more in context with source form indicated
Unclear if the depredation figures are only deriving from the surveyed households or from all households
Need to indicate when figures are ‘perceptions’ or ‘reported deaths’
Not clear reliability of results eg in number of ibex killed for trophy hunting – comment
It is important not only what trophy rates were but also how they were decided
Tourism is not discussed in a sophisticated manner – what is ‘mass tourism’ rate and how does this compare with other national parks? What facilities are there or what are tourists doing in the park? Is there waste at campsites? It is difficult to understand the type of tourist coming and activities undertaken in the Park and area, and also whether additional activities and economic opportunities have been started to provide local livelihood opportunities there
P13 check ‘above 90%’ – does not tally
Discussion
Discussion does not always fit results eg p 15. At times the discussion is more literature review and not critical analysis of the results of this paper. There is an interesting discussion of snow leopard conflict but it is not clear how much of the perception is from these communities. The discussion of tourism needs to be more critical and engaged in literature – normally national parks want tourists, but as more come there is a need to invest in more infrastructure eg toilets and accommodation, waste disposal and enforcement eg of speed limits. Yes, eco-tourism can help but there is much debate around what this means and how this might be perceived in relation to different contexts, especially when many tourists are local. Conclusions need to be reconceptualised and re written
Author Response
Reviewers' comments and Responses.
Reviewer #1:
Introduction Appropriate, but the detail of the National Park in footnote should be included in text in brief and offered in full in supplementary material
Response: Thank you for your comment. The description of the National Park in the footnote is included in text in brief. The full description is now corrected and incorporated in the text.
Reduce and make more concise the discussion of FGDs
Response: Thank you for your comment. The discussion on FGDs is reduced and concise now.
Comment on the all male groups
Response: Thank you for bringing up this point. According to the traditions of the study region, the male are responsible for management and decisions in daily life. The women are confined to the household activities only. The women have minimum role in the management outside home.
Could use some smaller quotes as well as longer paragraphs, and use more in context with source form indicated
Response: Thank you for your comment. The quotes were arranged and revised according. The context with source form indicated in the revision.
Unclear if the depredation figures are only deriving from the surveyed households or from all households
Response: Thank you for highlighting this mistake. These errors are now removed in the revised version.
Need to indicate when figures are ‘perceptions’ or ‘reported deaths’
Response: Thank you for identifying this mistake. The sentences are corrected now.
Not clear reliability of results eg in number of ibex killed for trophy hunting – comment
Response: Thank you for your comment. The results are very reliable and all the records about the trophy hunting is obtained from the KVO. These includes information about the name, passport number, permanent address, nationality of the trophy hunter etc, is obtained. While the information about the date of hunting, the size of the trophy horns, age and exact location of the hunt are also recorded. In addition, the notification order from the Wildlife Department for the trophy animal were also obtained. These orders have all the information about the trophy hunters and trophy animals. Moreover, the pictures available with the KVO including the Pakistani and International trophy hunters and the trophy animals were also obtained in soft. The KVO has also kept record of the porters carrying the materials with the trophy hunters for each hunt. These may be shared on demand.
It is important not only what trophy rates were but also how they were decided
Response: Thank you for guiding the clarity of this manuscript. It is already mentioned in the results. The KVO officials revealed in the FGD that no population census is carried out now. So according to their point of view trophy hunting is not a conservation tool now.
Tourism is not discussed in a sophisticated manner – what is ‘mass tourism’ rate and how does this compare with other national parks? What facilities are there or what are tourists doing in the park? Is there waste at campsites? It is difficult to understand the type of tourist coming and activities undertaken in the Park and area, and also whether additional activities and economic opportunities have been started to provide local livelihood opportunities there
Response: Thank you highlighting these issues. These issues are now discussed briefly. The activities of tourists, including rest, dining at the entrance of the park, the visiting spot of Pakistan China border is briefly discussed.
P13 check ‘above 90%’ – does not tally
Response: Thank you for your comment. The previous study mentioned above 90%, which is not actual. The calculated percent increase from the quota of 6 ibex to the trophy hunting of 11 ibex is 83.3%. This is now revised accordingly to show the actual figure.
Discussion does not always fit results eg p 15. At times the discussion is more literature review and not critical analysis of the results of this paper. There is an interesting discussion of snow leopard conflict but it is not clear how much of the perception is from these communities. The discussion of tourism needs to be more critical and engaged in literature – normally national parks want tourists, but as more come there is a need to invest in more infrastructure eg toilets and accommodation, waste disposal and enforcement eg of speed limits. Yes, eco-tourism can help but there is much debate around what this means and how this might be perceived in relation to different contexts, especially when many tourists are local. Conclusions need to be reconceptualised and re-written.
Response: Thank you for guiding the clarity of this manuscript. The discussion is improved now according to the suggestions.
We hope that this will be a final file. The uploaded file is having the Track Changes included to show the corrections made during the revision. Thank you so much for your kind comments.
Reviewer 2 Report
"Mass tourism and trophy hunting in and around KNP" is a very important contribution. It combines reliable statistics on the economic contributions of trophy hunting and tourism with collection of focus group data from key public and government stakeholders to paint a convincing picture of the effectiveness of local conservation efforts. The paper makes the crucial points that both trophy hunting and tourism can provide revenue to support conservation and local economies. But when trophy hunting is driven by demand rather than data-based population sustainability estimates, conservation efforts fail. The paper also notes that in the communities trophy hunting only improves public support for the trophy species themselves, and may reduce public support for predator species like wolves and snow leopards. Tourism appears to offer better prospects for sustainable conservation, but only if the environmental costs of tourism are controlled.
The authors do a good job of placing their work in the context of other studies that have also looked at balancing local economics, ethics, and conservation. Their methods are convincing. The footnote is probably not needed, and maybe should be included in supplementary material. The biggest deficiency of the paper is the presentation and labeling of graphs and tables. In Table 3, for example, two sets of means and standard deviations are included in columns on the right side, but I couldn't figure out what they referred to. The use of exponential notation in Fig. 4 A & B was also not helpful: conventional numbers, perhaps with an caption note that they indicate "in thousands of dollars" might be clearer. I assume that the formatting issues with Table 1 will be fixed before publication.
The manuscript would benefit from a good English language copy-edit.
Some minor points:
l. 90 Please also provide the altitude range for the study site.
l. 103. Please describe what KVO is when it's used the first time in the text.
l. 329-330. I think the authors are trying to say "90% above the allocated quota."
Author Response
Dear reviewer.
Many thanks for the efforts and detailed review. Please find a separate response to your kind comments below.
The footnote is probably not needed, and maybe should be included in supplementary material.
Response: Thanks for your comment. The footnote is shifted to removed and it is included in the text. The text is improved in the revised version.
In Table 3, for example, two sets of means and standard deviations are included in columns on the right side, but I couldn't figure out what they referred to.
Response: Thank you for guiding us to improve the paper. The Table 3 is revised now. The means and the standard deviations referred to the mean number of livestock and mean number of livestock depredated. The data was obtained from the surveyed households in the study area.
The use of exponential notation in Fig. 4 A & B was also not helpful: conventional numbers, perhaps with an caption note that they indicate "in thousands of dollars" might be clearer.Response: We appreciate the reviewer comment on this. We have added conventional numbers and revised the figures 4 (A & B).
I assume that the formatting issues with Table 1 will be fixed before publication.Response: Thank you for your comment. The formatting issues with Table 1 is fixed now.
The manuscript would benefit from a good English language copy-edit.Response: Thank you for identifying this deficiency. The manuscript is edited for the English language by native English speaker.
90. Please also provide the altitude range for the study site.Response: Thank you for your comment. The altitudinal range for the study site is now included in the article.
103. Please describe what KVO is when it's used the first time in the text.Response: Thank you highlighting this mistake. The KVO is described in the abstract where it is used for the first time.
329-330. I think the authors are trying to say "90% above the allocated quota."
Response: Thank you for highlighting this. In the previous study above 90% was mentioned which is not actual. The calculated percent increase from the quota of 6 ibex to the trophy hunting of 11 ibex is 83.3%. This is now revised accordingly to show the actual figure.
We wish to thank you for their detailed review, efforts and contribution to the scientific community as a reviewer.